# Prevalence of complementary and alternative medicine despite limited perceived efficacy in patients with rheumatic diseases in Mexico: Cross-sectional study

Claudia Isabel Caballero-Hernández[1], Susana Aideé González-Chávez[1], Adelfia Urenda-Quezada[2], Greta Cristina Reyes-Cordero[2], Ingris Peláez-Ballestas[3], Everardo Álvarez-Hernández[3], César Pacheco-Tena[1] *

1 Facultad de Medicina y Ciencias Biomédicas, Laboratorio PABIOM, Universidad Autónoma de Chihuahua, Chihuahua, México, 2 Facultad de Medicina y Ciencias Biomédicas, Universidad Autónoma de Chihuahua, Chihuahua, México, 3 Servicio de Reumatología, Hospital General de México "Dr. Eduardo Liceaga", México City, México

* dr.cesarpacheco@gmail.com

## Abstract

### Introduction

Complementary and alternative medicine (CAM) is frequently used by patients with rheumatic diseases (RD) to improve their symptoms; however, its diversity and availability have increased notably while scientific support for its effectiveness and adverse effects is still scarce.

### Objective

To describe the prevalence and diversity of CAM in patients with RD in Chihuahua, Mexico.

### Methods

A cross-sectional study was conducted in 500 patients with RD who were interviewed about the use of CAM to treat their disease. The interview included sociodemographic aspects, characteristics of the disease, as well as a description of CAM use, including type, frequency of use, perception of the benefit, communication with the rheumatologist, among others.

### Results

The prevalence of CAM use was reported by 59.2% of patients, which informed a total of 155 different therapies. The herbal CAM group was the most used (31.4%) and included more than 50 different therapies. The use of menthol-based and arnica ointments was highly prevalent (35%). Most patients (62.3%) reported very little or no improvement in their symptoms. Only a fourth of the patients informed the rheumatologist of the use of CAM. The use of CAM was influenced by female sex, university degree, diagnosis delay, lack adherence to the rheumatologist's treatment, family history of RD, and orthopedic devices.

**Data Availability Statement:** All relevant data are within the manuscript and its Supporting Information files.

**Funding:** The authors received no specific funding for this work.

**Competing interests:** The authors have declared that no competing interests exist.

## Conclusion

The use of CAM in our population is highly prevalent and similar to reports in different populations suggesting a widespread use in many different societies. We found high use of herbal remedies; however, there were many different types suggesting a lack of significant effect. Patients continue using CAM despite a perception of no-effectiveness. Recurrent use of CAM is explained by factors other than its efficacy.

## Introduction

Complementary and alternative medicine (CAM) is defined by the National Center for Complementary and Alternative Medicine (NCCAM) of National Institutes of Health (NIH) as a group of diverse medical and health care systems, practices, and products that are not presently considered to be part of conventional medicine [1]. According to the World Health Organization (WHO) the terms "complementary medicine" and "alternative medicine" refer to a broad set of health care practices that are not part of that country's own traditional or conventional medicine and are not fully integrated into the dominant health care system. In some countries these terms are used interchangeably with the term "Traditional Medicine," which the WHO defines as the sum of all the knowledge, skill, and practices based on the theories, beliefs, and experiences that are indigenous to different cultures, whether explicable or not, used in the maintenance of health, as well as in the prevention, diagnosis, improvement, or treatment of physical and mental illness [2].

The increased demand and use of CAM is a trend that began in the 1950s [3,4]. The factors that drive this trend have been analyzed in detail, and include several ideologic elements such as a holistic orientation to health disease treatment [5,6], desire for a more prominent role in disease treatment [7,8], the perception of being safer (natural) than conventional medical treatments [9–11], and openness to experience. Some factors also push patients away from conventional treatments toward CAM, such as dissatisfaction with the doctor-patient relationship, side effects from the treatment, poor disease response, and lack of access to conventional medical treatment. The WHO considers CAM to be an underestimated strategy, even with its widespread and growing use [12]. Despite its widespread use and popularity, several modalities of those CAM therapies that have been tested many lack comprehensive testing and few have failed to prove efficacy [13–23].

Patients with RD use CAM to control the pain or residual symptoms, and to deal with the side effects of standard treatments. In most cases, CAM is promoted and perceived as being based on natural components, and is safer and less toxic. This perception is strengthened in part by the continuous advertising of lifestyles that enhance the consumption of CAM products. These products represent a significant opportunity for profit because their manufacturing standards are less regulated than those of prescription drugs; and their real benefit and potential toxicity are uncertain; as a result, they are far easier to market. In some cases, patients treated whit CAM tend to relax their compliance with medical assessments and treatments because CAM empowers them to treat their disease [24], frequently in the wrong direction. CAM has been shown to delay the onset of disease-modifying antirheumatic drugs (DMARDs) in patients with rheumatoid arthritis (RA), and can be related to their withdrawal [25,26]; however, higher compliance has been reported in users of CAM in Chinese-American rheumatic patients [27]. It has also been reported that patients' perception of underperformance of conventional treatments is not necessarily confirmed by objective measures [28].

The accessibility, diversity, and demand for CAM have increased, and several factors have been proposed to explain this change. The economic value of the market of products has risen significantly, and widespread and effective advertising has transformed CAM into a cultural trend and lifestyle. However, CAM has potential risks, some of which are as a result of the therapy itself or by the interaction with formal medical treatment.

Previous reports have shown that 60%–85% of Mexican patients with RD use CAM [29–33]; however, the types and availability of CAM treatments have changed. Recently, the perception of risk associated with the use of CAM was evaluated in 246 patients with RA from Latin American countries, including Mexico, [32] and it was found that 81.3% were CAM users and 28.5% had a significant risk perception.

CAM is indeed a relevant element in the treatment of RD, and is driven by local factors, which continuously evolve as new CAM modalities appear and become popular. We consider that a better understanding of the trends in CAM use could anticipate the spread of risky alternatives and allow strategies to accomplish higher compliance of the patients to their standard therapy, in the understanding that the final decision in regard to CAM use relies in the patients' will. As the diversity and availability of CAM treatments have not been explored in our population, we aimed to evaluate the frequency, diversity, and factors associated with the use of CAM in patients with RD in Chihuahua, Mexico. This research was intended to be as exhaustive as possible and include as many different treatments to serve as a potential reference to understand actual trends in CAM use and selection.

## Methods

### Study design

A cross-sectional study was conducted in patients with RD from six public care clinics in Chihuahua, Mexico, from January to April 2019.

### Participants

The study included patients with a previous diagnosis of RD according to standard classification criteria [26–29] who were in the waiting room for their rheumatology checkup. All of the patients were previously known and diagnosed by the referring physicians (CPT, GRC, AUQ), and were under treatment and follow-up; there were no external referrals, and patients whose cognitive and verbal capacity allowed them to provide informed consent to participate in the study were included. The exclusion criteria included those patients who expressed unwillingness to answer the questionnaire, or those who did not complete the interview for any other reason.

### Variables and interview instruments

The questionnaires was applied by trained interviewer (ICH) face-to-face in Spanish, independently in the clinics they attended, to collect the following variables: 1) sociodemographic: age, sex, marital status, occupation, and education level; 2) related to RD: type of RD, disease duration, family history of RD, self-report of disease activity and adherence to rheumatological treatment, use of orthopedic device, among others; and 3) CAM use: type, frequency of use, reason for use, perceived improvement, sources of information channel, and physician-related aspects of CAM. The interviewer had no previous connection to the patients who were interviewed. Each patient underwent one interview, which lasted less than 30 min in all cases.

Disease activity was self-reported using the Routine Assessment of Patients Index Score-3 (RAPID-3) previously validated in the Spanish language [34–37], which is a pooled index for

function, pain, and patient global estimate of status. Each of the three individual measures was scored from 0 to 10, for a total of 30. Based on RAPID-3 scores, disease severity was classified as high (>12), moderate (6.1–12), low (3.1–6), or remission (= 3).

The 4-item Morisky Medication Adherence Scale (MMAS-4) was used to evaluate the adherence to rheumatological treatment [38–40]. The MMAS-4 has four questions, each with a binary answer that allows patients to be classified as adherent or non-adherent to their rheumatological treatment.

The description of CAM use was conducted using a questionnaire that included the type, frequency of use, perception of the use benefit, sources of information, and communication with the doctor. As the variety of CAM in each region is different, an exploratory phase was performed to identify the main CAM therapies offered to our population for the relief of RDs and their symptoms, and a visual catalog was built to be shown to patients during interviews. For this, visits to the main establishments that offer CAM in Chihuahua were made, and a search on social networks and digital media was performed to expand the catalog. The initial image catalog was shown to 15 patients who were asked to indicate which of these products they recognized (whether they had used them or not), and the patients were also asked if they knew of any other product or service that was not included in the presentation shown. Additionally, in order to find information on providers of Traditional Indigenous Medicine, the "Coordinadora Estatal de la Tarahumara," whose function is to contribute to the development of the indigenous people of the region, was contacted. Likewise, the Festival of Crafts of Urban Indigenous Communities was attended, with the intention of expanding the catalog. Images of each of the products were obtained, and a visual catalog of 54 products and 27 therapies was built. The image catalog and the questionnaire were reviewed by three expert rheumatologists, and necessary modifications were made. The visual catalog of CAM products and therapies included 155 different CAM treatments, which were classified into seven groups as follows: herbal, oral supplements, ointments/oils, mind and body practices, food-based, energy field manipulation, and others.

Additional items were included regarding the physician's attitude toward the use of CAM, patient-physician communication, and the overall functional conditions of the patient, including the use of orthopedic devices (such as a cane or walker) were evaluated.

## Pilot study

To evaluate the content of the questionnaire, the duration of the interviews, the clarity of the questions, the disposition of the patients, the time available in the waiting room prior to their medical appointment, and the clarity of the instructions for the interviewer, a pilot test was performed in two of the clinics included in the study. The pilot study included 22 patients, and functioned to improve the content of the graphical catalog of diverse CAM. We considered these opinions as a validation for face validity given that no further recommendations were given. The questionnaires were not given to the patient; instead, the interviewer asked every question and recorded the answer. After validation results were analyzed by the multidisciplinary group, modifications were made to create the final version of the questionnaire and the visual catalog.

## Sample size

The sample size was calculated based on a finite population approach. We previously reported a prevalence of 21.4% for RD in our population (Chihuahua, Mexico) [41], and the most recent census showed a total population of 878,062 in the city (2015). Therefore, our universe included 187,905 potential patients, with a confidence interval (CI) of 95% and an alpha of

0.05. We obtained a sample size of 383 patients; however, we included 500 patients to better assess the diversity of CAM. Patients were included in the consecutive sampling.

## Statistical analysis

Based on the interviews, a database was generated in which the variables were coded by three of the participating researchers (ICH, SGC, CPT), and were defined as ordinal, nominal, or categorical. For logistic regression analysis, the variables were dichotomized.

The absolute and relative frequencies of ordinal, nominal, or categorical variables were used. A descriptive analysis was performed with measures of central tendency and dispersion for continuous variables and mean ± standard deviation (SD). To determine the differences in the variables between CAM users and non-users, the χ2 test was used for categorical variables, the Student's t-test was used for continuous variables, and binary and multivariable logistic regressions were used to investigate factors associated with CAM use in patients with RDs. Odds ratios (ORs) and 95% CIs were determined. Continuous variables, including age, disease duration, diagnosis delay, and RAPID-3, were dichotomized considering their average as the cut-off point, while the variable occupation was dichotomized by classification as high mechanical demand (blue-collar workers) or low mechanical demand (white-collar workers). Variables with a p-value $< 0.2$ in the binary analysis were included in the multivariable logistic regression analysis. Considering the inequality between the proportion of each RD, in which patients with RA accounted for 73% of the population in our study, the variables corresponding to RD were not included in the multivariate analysis, even though their p-value was $< 0.2$. The Hosmer–Lemeshow goodness-of-fit test for the multivariable model yielded a chi-square of 6.7 (p = *0.460*). SPSS version v24.0 (IBM) was used for the statistical analysis. Statistical significance was set at p $< 0.05$.

## Ethical approval

This study was approved by the Ethical Committee of the Faculty of Medicine and Biomedical Sciences of the Autonomous University of Chihuahua (RI-019-19). Patients who agreed to participate signed an informed consent form.

## Results

Five hundred consecutive patients with RD were included, and none declined the interview. The patients were predominantly female (81.2%), and their average age was 50.36 ± 14.58 years. Most of the patients were Mexican mestizos (97.4%). RA was the most prevalent disease in patients, followed by ankylosing spondylitis (AS) and systemic lupus erythematosus (SLE) (Table 1). Most patients (61.9%) were diagnosed within the first 2 years of disease onset. Disease activity measured with the RAPID-3 questionnaire was used to classify the patients as high (39.8%), moderate (21.6%), and low (16.4%) disease activity, or remission (22.2%), while the MMAS-4 showed that 50.2% of the patients adhere to rheumatological treatment. The most common comorbidities were arterial hypertension (43.4%), diabetes mellitus (17.4%), and hypothyroidism (15.3%).

Two hundred and ninety-six patients (59.2%) reported using CAM. Patients with fibromyalgia, psoriatic arthritis (PsA), and RA had the greatest use of CAM (100%, 70.5%, and 61.9%, respectively), while patients with SLE, osteoarthritis (OA), and AS used CAM to a lesser extent (47.5%, 44.4%, and 42.8%, respectively). No significant differences were found among these proportions (p = 0.088) in the overall comparison. The prevalence of CAM use was higher in women (84.1% vs. 76.9%, p = 0.05) and in patients with a university education level (22.0% vs. 15.7%, p = 0.05). Meanwhile, occupation, marital status, and type of medical coverage were

**Table 1. Sociodemographic data.**

| Variable | All the patients n = 500 | CAM users n = 296 | CAM non-users n = 204 | *p* |
|---|---|---|---|---|
| Age (mean ± SD) | 50.36 ± 14.58 | 50.7±14.2 | 49.87±15.1 | *0.72[a]* |
| Sex (women/men) | 406/94 | 249/47 | 157/47 | ***0.05*** [b] |
| Marital status | | | | |
| Married (%) | 56.8 | 57.8 | 55.4 | *0.42* [b] |
| Single (%) | 43.2 | 42.2 | 44.6 | *0.23* [b] |
| Occupation | | | | |
| Home (%) | 48.2 | 52.7 | 41.6 | *0.35* [b] |
| Office work (%) | 28.0 | 28.4 | 27.4 | *0.82* [b] |
| Construction (%) | 11.6 | 10.1 | 13.7 | *0.70* [b] |
| Student (%) | 3.6 | 2.3 | 4.9 | *0.19* [b] |
| Farmer (%) | 1.8 | 1.0 | 2.9 | *0.34* [b] |
| Unemployed (%) | 3.2 | 1.6 | 5.3 | *0.23* [b] |
| Retired (%) | 2.0 | 2.02 | 1.96 | *0.77* [b] |
| Education level | | | | |
| Elementary (%) | 24.0 | 22.6 | 25.9 | *0.22* [b] |
| Junior High (%) | 31.0 | 32.1 | 29.4 | *0.29* [b] |
| High School (%) | 21.4 | 19.6 | 24.0 | *0.14* [b] |
| University (%) | 19.4 | 22.0 | 15.7 | ***0.05*** [b] |
| Postgraduate (%) | 1.6 | 1.7 | 1.5 | *0.58* [b] |
| None (%) | 2.6 | 2.0 | 3.4 | *0.24* [b] |
| Rheumatic disease | | | | |
| Rheumatoid arthritis (%) | 73.0 | 76.4 | 68.1 | ***0.03*** [b] |
| Ankylosing Spondylitis (%) | 9.8 | 7.4 | 13.2 | ***0.02*** [b] |
| Lupus (%) | 8.0 | 6.4 | 10.3 | *0.08* [b] |
| Psoriatic arthritis (%) | 3.4 | 4.1 | 2.5 | *0.23* [b] |
| Osteoarthritis (%) | 1.8 | 1.4 | 2.5 | *0.28* [b] |
| Fibromyalgia (%) | 1.4 | 2.4 | 0 | ***0.02*** [b] |
| Other (%) | 2.6 | | | *0.70* [b] |
| Disease duration (years) (%) | 12.5 ± 10.2 | 13.8 ± 10.8 | 10.6 ± 8.8 | ***0.001[a]*** |
| Diagnosis delay (years) (%) | 2.5 ± 5.3 | 2.9 ± 6.1 | 1.9 ± 3.6 | ***0.001[a]*** |
| RAPID-3 (mean ± SD) | 10.5 ± 7.5 | 11.33 ± 7.5 | 9.2 ± 7.2 | ***0.001[aa]*** |
| Treatment adherent (MMAS-4) (%) | 50.2 | 46.3 | 55.9 | ***0.037*** [b] |
| Family history of RD (%) | 47.8 | 51.4 | 42.6 | |
| Use of orthopedic devices (%) | 14.2 | 19.7 | 6.4 | *<**0.001*** [b] |
| Requires caregiver (%) | 35.6 | 40.2 | 28.9 | ***0.006*** [b] |
| Forgets to take medicines (%) | 37.0 | 38.5 | 34.8 | *0.227* [b] |
| Stop using treatment if improves (%) | 13.0 | 14.9 | 10.3 | *0.086* [b] |

[a] t-student test

[b] χ2.

MMAS-4: 4-item Morisky's Medication Adherence Scale; RAPID-3: Routine Assessment of Patients Index Score-3.

not significantly different between CAM users and non-users (Table 1). A higher proportion of patients with RA and fibromyalgia were CAM users (p = 0.03 and p = 0.02, respectively), while patients with AS generally did not use CAM (p = 0.02). The comparisons of these proportions were not significantly different for the rest of the disease (Table 1). CAM users had a longer disease duration (p = 0.001), diagnosis delay (p = 0.001), and higher disease activity

according to RAPID-3 (p = 0.001), and lower adherence to rheumatological treatment (p = 0.037) than non-CAM users (Table 1).

Of the CAM groups, herbal medicine was the most frequently used, followed by ointments/oil, oral supplements, mind and body practices, others (including homeopathic medicine), and energy fields manipulation (Fig 1A). The five most common CAM treatments were

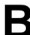

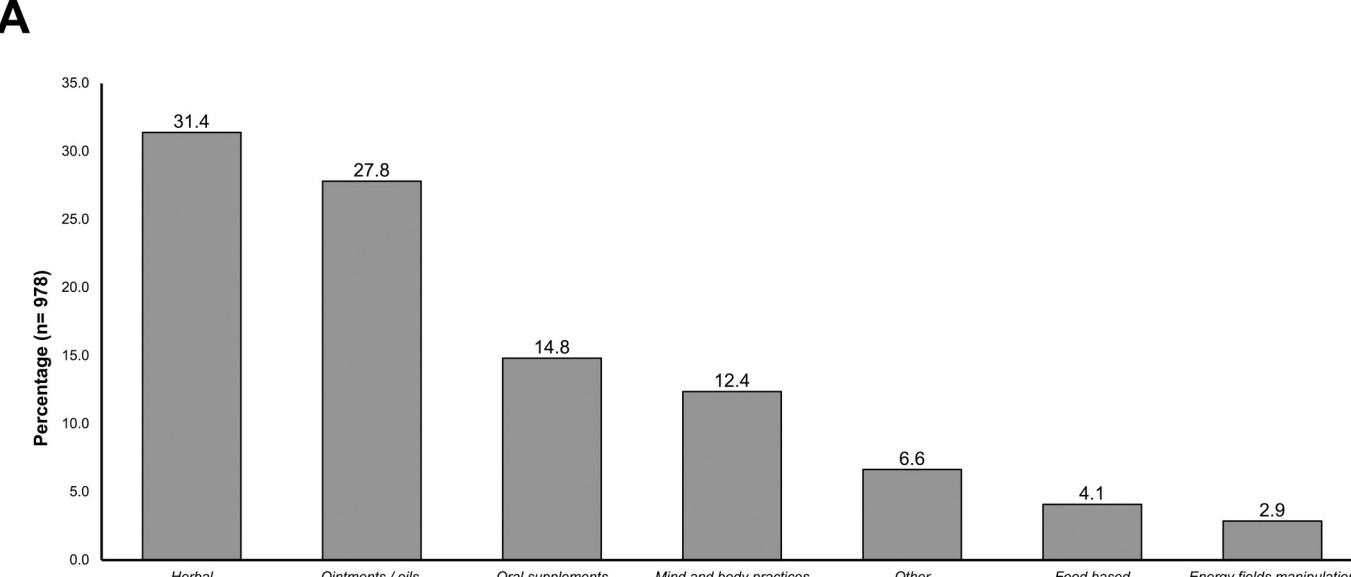

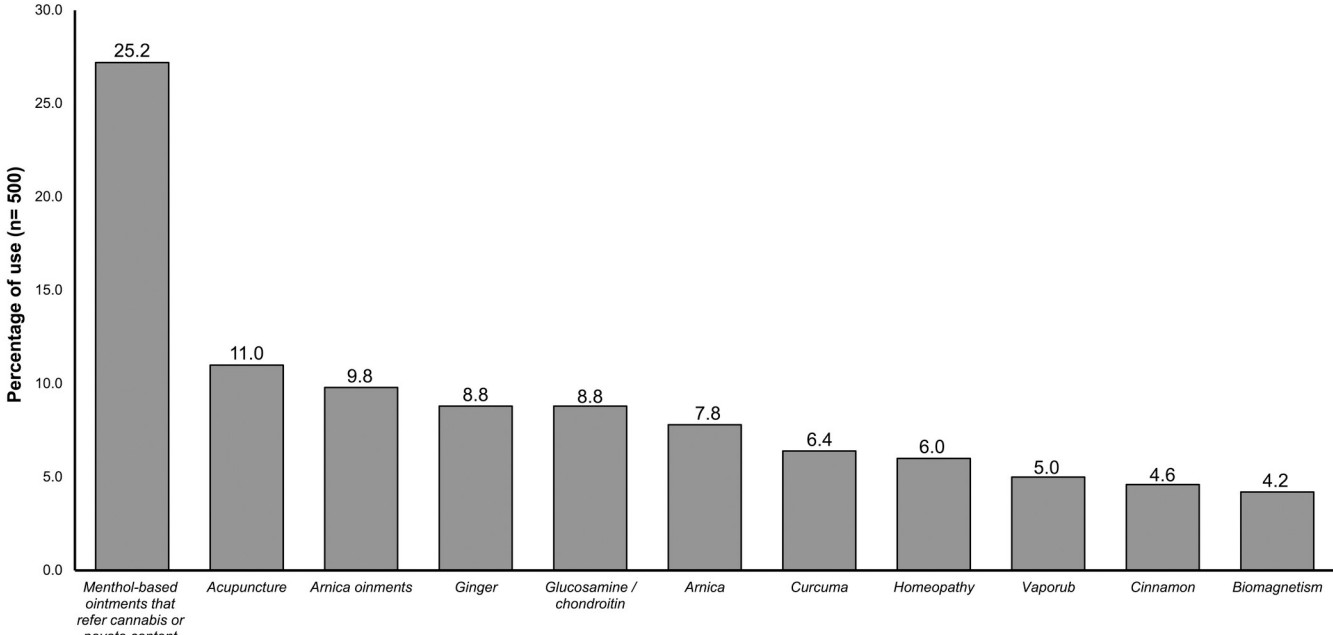

**Fig 1. Prevalence of use of CAM therapies in patients with rheumatic diseases.** (A) The percentages of CAM use classified in the seven groups are shown; the total number of different therapies reported was 978, the bars indicate the proportion corresponding to each group. (B) The percentages of use of the eleven most used individual CAM therapies in the population (n = 500) are shown.

menthol-based ointments that reported cannabis or peyote (*Lophophora williamsii*) content, acupuncture, arnica ointments, ginger, and glucosamine supplements (Fig 1B).

A complete list of 155 different therapies belonging to each group is presented in Table 2. In the herbal CAM group, extracts of ginger, arnica, and turmeric were the most common, while cinnamon and chamomile (mostly as tea infusions) were also considered therapeutics. In a long list, different options were used by < 5% of the sample, showing the lack of a clear trend for herbal remedies in our society, despite the specific recommendations for the herb purposes by distributing stores. Oral supplements were used by 14.8% of the patients, the most frequently of which was glucosamine, either with or without chondroitin, followed by fish oil capsules. A large variety of oral supplements were used by a minority of patients, suggesting the absence of a trend. These supplements include transfer factor, Immunocal®, collagen, shark cartilage, and commercial well-being supplements (Omnilife® and Herbalife®). A hundred and fifty patients reported the use of cannabis-related CAM (Table 3), mostly menthol-based ointments that presumably contain cannabis and/or peyote as the label or product name implies. However, only eight patients reported smoking cannabis and six consumed it orally, either in capsules, drops, or infusions. We found no significant trend toward the use of synthetic cannabinoids.

The patients reported using an average of 3.3 ± 3.0 different CAM therapies, spending an average of $19.6 ± 44.4 USD per month. CAM therapies were used by 64.4% of the population ≥ 15 days per month, and 61.1% of the population used CAM therapies for more than a year. Only 12 patients reported adverse reactions, none of which were severe (pain, burning, or rash). The main reason for CAM use was to treat joint pain, followed by inflammation, and a minority used CAM to improve their overall well-being, or "cure" the disease (Table 4). Only a minority expected CAM to cure the disease, and in most cases, the participants considered CAM as a palliative alternative. Moreover, 62% of the patients reported very little or no improvement in their symptoms.

Only a quarter of the patients informed their physician of their use of CAM, and a minority (13.7%) indicated that the physician specifically asked them about their use of CAM. The physician's attitude regarding the use of CAM (as interpreted by the patient) was variable, ranging from agreement to opposition, and a proportion showed a neutral position (Table 5).

The variables with p < 0.20 in the binary logistic regression analysis (Table 6) were included in the multivariate analysis (Table 7). The results of multivariate analysis showed that the use of CAM was significantly associated by the female sex (OR, 1.79; 95% CI, 1.1–2.9), university education (OR, 0.55; 95% CI, 0.34–0.90), diagnosis delay (OR, 0.52; 95% CI, 0.33–0.82), adherence to treatment (OR, 0.66; 95% CI, 0.45–0.97), family history of RD (OR, 1.54; 95% CI, 1.01–2.6), and the use of orthopedic devices (OR, 0.281; 95% CI, 0.15–0.54).

## Discussion

The present study describes the prevalence and diversity of CAM in patients with RD in Chihuahua, Mexico, and shows that over 50% of patients use at least one CAM treatment, mostly to improve the symptoms of the disease. Although patients with RD use CAM, a significant proportion perceive little or no benefit. Herbal CAM modalities were the most varied and frequent in their use, while menthol-based ointments considered to include cannabis or peyote were the most commonly used. Multivariate analysis identified several factors that associated the use of CAM, including female sex, having a university degree, delay in the diagnosis of disease, lack of compliance with the rheumatologist's treatment, family history of RD, and use of orthopedic devices. Several other factors, such as age, religion, labor, and marital status did not

**Table 2. CAM description by group and frequency of use.**

| Herbal—frequency (%) | | | |
|---|---|---|---|
| Ginger– 44 (8.8) | Atridol * – 7 (1.4) | Bay leaves– 2 (0.4) | Orange bloosom– 1 (0.2) |
| Arnica– 39 (7.8) | Gobernadora– 5 (1.0) | Chivo pez– 2 (0.4) | Black pepper– 1 (0.2) |
| Turmeric– 32 (6.4) | Boldo– 4 (0.8) | Eucalyptus– 2 (0.4) | Parsley– 1 (0.2) |
| Cinnamon– 23 (4.6) | Tizana uva– 4 (0.8) | Mesquite– 1 (0.2) | Celery– 1 (0.2) |
| Cannabis– 14 (2.8) | Rosemary– 4 (0.8) | Salvia– 1 (0.2) | Cat's claw– 1 (0.2) |
| Chamomile– 13 (2.6) | Shave grass– 4 (0.8) | Chaya– 1 (0.2) | Ruda– 1 (0.2) |
| Moringa– 12 (2.4) | Peppermint– 3 (0.6) | Eryngo– 1 (0.2) | Green tea– 1 (0.2) |
| Palo Azul– 11 (2.2) | Thyme– 3 (0.6) | Orange leaf– 1 (0.2) | Goji– 1 (0.2) |
| Nettle– 11 (2.2) | Soursop– 3 (0.6) | Del pasmo– 1 (0.2) | Herbal– 1 (0.2) |
| Seven flowers– 9 (1.8) | Taheebo– 2 (0.4) | Hierba del peru– 1 (0.2) | Chia– 1 (0.2) |
| Linseed– 9 (1.8) | Cayenne pepper– 2 (0.4) | Epazote– 1 (0.2) | Dandelion– 1 (0.2) |
| Mullein flowers– 9 (1.8) | Stramonium– 2 (0.4) | Flor de peña– 1 (0.2) | Spirulina– 1 (0.2) |
| Osha– 7 (1.4) | Elder flower– 2 (0.4) | Jamaica– 1 (0.2) | Belladonna– 1 (0.2) |
| Oral supplements—frequency (%) | | | |
| Glucosamine– 44 (8.8) | Shark cartilage– 5 (1.0) | Silipharma– 2 (0.4) | Oxivit– 1 (0.2) |
| Fish oil– 11 (2.2) | Xi ac ** – 5 (1.0) | Stem cells– 2 (0.4) | Butanoic acid– 1 (0.2) |
| Transfer factor– 10 (2.0) | Omnilife– 3 (0.6) | Scorpion poison– 1 (0.2) | Alkaline water– 1 (0.2) |
| Magnesium chloride– 10 (2.0) | Rhus toxicendron– 3 (0.6) | Neo vita– 1 (0.2) | Amway– 1 (0.2) |
| Immunocal– 9 (1.8) | Mineral serums– 3 (0.6) | Sea water– 1 (0.2) | Vitamin C– 1 (0.2) |
| Collagen– 6 (1.2) | DoXi ** – 3 (0.6) | Rheumacol– 1 (0.2) | Iso-xp– 1 (0.2) |
| Noni juice– 6 (1.2) | Colloidal silver– 3 (0.6) | Artribion– 1 (0.2) | |
| Herbalife– 6 (1.2) | Herbasan– 2 (0.4) | GH3–1 (0.2) | |
| Ointments/Oils—frequency (%) | | | |
| Menthol-based that refers cannabis or peyote content– 126 (25.2) | Roble oil– 4 (0.8) | Frescapiel– 2 (0.4) | Avocado/ocote– 1 (0.2) |
| | Maravi– 3 (0.6) | Miracle ointment– 2 (0.4) | Spray El jorobadito– 1 (0.2) |
| | Olive oil– 3 (0.6) | Alcohol- peyote– 1 (0.2) | Petroleum– 1 (0.2) |
| Arnica– 49 (9.8) | Arthrostop cream– 3 (0.6) | Rattlesnake– 1 (0.2) | WD-40–1 (0.2) |
| VapoRub– 25 (5.0) | Cannabis oil– 2 (0.4) | Aluminium– 1 (0.2) | Pinol cleaner– 1 (0.2) |
| Balsamo del tigre– 12 (2.4) | Peyote oil– 2 (0.4) | Sauce cream– 1 (0.2) | Green alcohol– 1 (0.2) |
| Mamisan– 10 (2.0) | Hyaluronic acid– 2 (0.4) | Menthol– 1 (0.2) | Dr. Bell's– 1 (0.2) |
| Alcohol-cannabis– 5 (1.0) | Seven flowers– 2 (0.4) | Bengue– 1 (0.2) | |
| Coyote bait– 4 (0.8) | Viejito ointment– 2 (0.4) | Argan oil– 1 (0.2) | |
| Mind and body practices—frequency (%) | | | |
| Acupuncture– 55 (11.0) | Thermal baths– 5 (1.0) | Dancing– 2 (0.4) | Wood therapy– 1 (0.2) |
| Iridology– 11 (2.2) | Reiki– 5 (1.0) | Hypnosis– 2 (0.4) | Aromatherapy– 1 (0.2) |
| Chiropractor– 9 (1.8) | Atriotherapy– 4 (0.8) | Acupressure– 2 (0.4) | Sand theraphy– 1 (0.2) |
| Shaman– 6 (1.2) | Reflexology– 4 (0.8) | Hydrotherapy– 2 (0.4) | |
| Massotherapy– 6 (1.2) | Ozone therapy– 3 (0.6) | Yoga– 2 (0.4) | |
| Food based—frequency (%) | | | |
| Apple vinegar– 12 (2.2) | Nopal– 3 (0.6) | Grenetin– 1 (0.2) | Sotol/onion/garlic– 1 (0.2) |
| Garlic– 11 (2.2) | Purple onion– 2 (0.4) | Chickpea– 1 (0.2) | Honey– 1 (0.2) |
| Aloe Vera– 4 (0.8) | Papaya juice– 2 (0.4) | Pineapple– 1 (0.2) | Oats– 1 (0.2) |
| Energy field manipulation—frequency (%) | | | |
| Biomagnetism– 21 (4.2) | Electrotherapy– 2 (0.4) | Magnesium Bracelet– 1 (0.2) | Pellets– 1 (0.2) |
| Cooper bracelet– 3 (0.6) | | | |
| Others—frequency (%) | | | |
| Homeopathy– 30 (6.0) | Naturist– 11 (2.2) | Platelet rich plasma– 3 (0.6) | Natural vaccine– 1 (0.2) |

(*Continued*)

**Table 2.** (Continued)

| | | | |
|---|---|---|---|
| Apitherapy– 12 (2.2) | Urine therapy– 8 (1.6) | | |

* Herb mix

** Sold as natural but have been shown to have corticosteroids.

predict the use of CAM; indeed, patients with higher education were more prone to use CAM (mostly oral supplements).

We found that 59.2% of the population indicated that they were currently or had previously used CAM, which is lower than previous reports in the Mexican population [29–33], but similar to others in different populations [25,27,42–44]. The prevalence and types of CAM in patients with RD varies among different countries, and a trend for the use of oral supplements is observed. Oral supplements are the most common type of CAM reported in Australia [42], Saudi Arabia [43], and Japan [44], and generally include vitamins, fish oils, and nutritional supplements; in most cases, their safety profile is known. Several predictors of CAM use have been described; generally, women use CAM more than men [27,42–44]. In our population, women used more CAM more frequently than men, but this was not significant in the multivariate analysis.

Herbal CAM was the most common group in our study. The most commonly used herbs are traditional remedies from our local culture, and on a lesser scale, some Chinese herbal remedies. Ointment, which presumably contains cannabis and/or peyote, are widely available and were commonly used in our population, although it is unclear whether the ointments actually contain cannabis. Other formulations of cannabis (including the direct use of the plant) were only used by a minority of participants, and only for short periods. Herbal remedies are potentially inconsistent in their preparation and have an unattainable distribution [48]. They are mostly self-administered, and frequently treating physicians unaware that the patients are using them.

Our results show that a high proportion of patients who used CAM did not improve, even after testing several modalities; this recurrent use has been reported and is based on personal philosophical perceptions [9]. Indeed, religious, political, and philosophical positions and

**Table 3. Use of cannabis-related CAM.**

| Variable | Frequency (%) (n = 150) |
|---|---|
| Administration | |
| Topical | 136 (90.7) |
| Smoked | 8 (5.3) |
| Oral | 6 (4.0) |
| Type | |
| Menthol-based ointments that refer cannabis or peyote content | 126 (84.0) |
| Smoked cannabis | 8 (5.3) |
| Cannabis alcohol-based ointment | 5 (3.3) |
| Cannabis infusion | 4 (2.7) |
| Cannabis oil | 2 (1.3) |
| Peyote oil | 2 (1.3) |
| THC drops | 1 (0.7) |
| Cannabis capsules | 1 (0.7) |
| Peyote alcohol-based ointment | 1 (0.7) |

Table 4. Patient rationale on the use and effect of CAM.

| Variable | | All CAM | Herbal | Oral suppl. | Ointments/oils | Mind and body practices | Food based | Energy field manipulation | Other |
|---|---|---|---|---|---|---|---|---|---|
| **Reason of use** | *Joint pain* | 79.8 | 84.6 | 80.0 | 86.3 | 81.0 | 80.0 | 89.7 | 63.6 |
| | *Joint swelling* | 10.6 | 11.6 | 2.7 | 7.9 | 5.8 | 15.0 | 0 | 7.6 |
| | *Cure the disease* | 4.8 | 0.3 | 8.7 | 0 | 9.9 | 2.5 | 6.9 | 24.2 |
| | *Improve immunity* | 2.7 | 2.7 | 8.0 | 2.2 | 3.3 | 0 | 3.4 | 4.5 |
| | *Other* | 2.1 | 0.7 | 0.7 | 3.6 | 0 | 2.5 | 0 | 0 |
| **Perceived improvement** | *Same as before* | 34.9 | 44.7 | 40.7 | 35.3 | 52.9 | 35.0 | 44.8 | 65.5 |
| | *Little* | 27.4 | 24.6 | 20.0 | 30.9 | 15.7 | 25.0 | 20.7 | 15.3 |
| | *Good* | 22.3 | 20.5 | 18.0 | 18.0 | 14.0 | 30.0 | 13.8 | 12.1 |
| | *Very good* | 15.4 | 10.2 | 21.3 | 15.8 | 17.4 | 10.0 | 20.7 | 7.6 |
| **Information channel** | *Family* | 70.5 | 75.1 | 62.0 | 77.0 | 66.9 | 62.5 | 69.0 | 78.8 |
| | *Friend* | 14.0 | 10.6 | 15.3 | 8.6 | 26.4 | 17.5 | 13.8 | 13.6 |
| | *TV* | 6.2 | 1.0 | 5.3 | 5.8 | 0.8 | 5.0 | 0 | 4.5 |
| | *Internet* | 4.5 | 10.2 | 8.0 | 3.6 | 2.5 | 10.0 | 10.3 | 1.5 |
| | *Journal* | 2.7 | 1.7 | 5.3 | 2.9 | 1.7 | 15.0 | 3.4 | 1.5 |
| | *Doctor* | 2.1 | 1.4 | 4.0 | 2.2 | 1.7 | 0 | 3.4 | 0 |

specific health perceptions have been applied to profile the use of CAM [45]. The lack of significant effects of several CAM modalities to improve the symptoms of RD has been reported previously. Interestingly, a systematic review of 18 randomized controlled trials of CAM in RA, including herbal and homeopathy, showed that the effects of CAM were either non-superior to the placebo or very limited in a significant outcome, as determined by the health assessment questionary (HAQ) or swollen joint count. In addition, CAM is inferior to methotrexate. This systematic review suggests that CAM, as a best case scenario, can be used as complementary measure to traditional treatment, but is not free from adverse events [15]. Another systematic

Table 5. Physician related aspects of CAM.

| Variable | Frequency (%) n = 296 |
|---|---|
| Perception of the most reliable type of therapy | |
| Treatment provided by rheumatologist | 215 (72.6) |
| Complementary and Alternative medicine | 4 (1.4) |
| Both therapies | 77 (26.0) |
| Perception of disease control with rheumatological treatment | 248 (83.8) |
| Report CAM use to rheumatologist | 73 (24.7) |
| Reason why the rheumatologist was informed of the use of CAM | |
| "The doctor must know everything I'm taking" | 231 (78.1) |
| "The doctor asked me" | 41 (13.7) |
| "Know if you have any interactions with the drug" | 16 (5.5) |
| "Ask how it works" | 8 (2.7) |
| Rheumatologist's position regarding the use of CAM | |
| Disagreement | 103 (34.8) |
| Agree | 91 (30.7) |
| Indifferent | 102 (34.5) |
| Advice from the CAM practitioner to withdraw rheumatologist treatment | 46 (15.5) |
| Suspension of rheumatological treatment when using CAM | 35 (11.9) |

**Table 6. Predictive factors by binary logistic regression analysis for CAM use in Mexican patients with RD (n = 500).**

| Variable | OR | 95% CI | p |
|---|---|---|---|
| Age (>50 years) | 1.06 | 0.72–1.52 | *0.40* |
| Sex (female) | 1.59 | 1.01–2.49 | ***0.05*** |
| Occupation (white collar) | 1.14 | 0.75–1.72 | *0.30* |
| Education level | | | |
| Elementary | 0.83 | 0.55–1.20 | *0.22* |
| Junior High | 0.88 | 0.59–1.29 | *0.29* |
| High School | 1.30 | 0.84–1–99 | *0.14* |
| University | 1.51 | 0.95–2.40 | ***0.05*** |
| Postgraduate | 0.86 | 0.20–3.66 | *0.58* |
| None | 1.72 | 0.57–5.18 | *0.24* |
| Rheumatic disease | | | |
| Rheumatoid arthritis | 1.51 | 1.014–2.24 | ***0.03*** |
| Ankylosing Spondylitis | 0.52 | 0.29–0.95 | ***0.02*** |
| Lupus | 0.60 | 0.31–1.14 | ***0.08*** |
| Psoriatic arthritis | 1.67 | 0.58–4.84 | *0.23* |
| Osteoarthritis | 0.54 | 0.14–2.05 | *0.28* |
| Disease duration (longer) | 1.75 | 1.18–2.54 | ***0.002*** |
| Diagnosis delay (years) | 0.49 | 0.32–0.77 | ***0.001*** |
| RAPID-3 (lower) | 0.663 | 0.46–0.95 | ***0.016*** |
| Treatment adherent (MMAS-4) | 0.68 | 0.48–0.98 | ***0.022*** |
| Family history of RD | 1.42 | 0.99–2.03 | ***0.034*** |
| Use of orthopedic devices | 3.60 | 1.91–6.75 | ***<0.001*** |
| Requires caregiver | 1.65 | 1.12–2.42 | ***0.006*** |

Continuous variables were dichotomized. Age: < 50.36 years/> 50.36 years; occupation: Blue collar workers/white collar workers; Disease duration: < 12.50 years/> 12.50 years; Diagnosis delay: < 2.53 years/> 2.53 years; RAPID-3: < 10.47/> 10.47. The $\chi^2$ test was used to determine statistical significance.

MMAS-4: 4-item Morisky's Medication Adherence Scale; RD: Rheumatic disease; RAPID-3: Routine Assessment of Patients Index Score-3.

review of the effectiveness of CAM in RA showed inconsistencies between the different modalities [14].

If a marginal effect is accepted and no attempt is made to substitute the standard of care guidelines, several CAM modalities have shown a certain degree of benefit, including nutritional strategies using fish oil, vitamin D, and probiotics; likewise, some herbal alternatives are

**Table 7. Predictive factors by multivariate logistic regression of CAM use for patients with RD (n = 500).**

| Variable | B | OR | 95% CI | p |
|---|---|---|---|---|
| Sex | 0.58 | 1.79 | 1.10–2.90 | ***0.019*** |
| University degree | -0.60 | 0.55 | 0.34–0.90 | ***0.017*** |
| Diagnosis delay | -0.66 | 0.52 | 0.33–0.82 | ***0.005*** |
| Treatment adherent (MMAS-4) | -0.41 | 0.66 | 0.46–0.97 | ***0.032*** |
| Family history of RD | 0.43 | 1.54 | 1.06–2.57 | ***0.025*** |
| Use of orthopedic devices | -1.27 | 0.28 | 0.15–0.54 | ***<0.001*** |

Chi-square goodness-of-fit = 6.7 (p = 0.460). MMAS-4: 8-item Morisky's Medication Adherence Scale.

currently being explored for their use in RA [14]. Similarly, in other fields of non-pharmacological treatment of RD, such as physiotherapy, psychotherapy, balneology, and rehabilitation, limited access to research funding for the effectiveness of CAM has led to a slow increase in scientific evidence. However, recent publications have confirmed the interest in CAM as a research field, particularly in China, with most research dealing with herbal or dietary supplements [46,47].

In the present study, the primary recommendation source for the use of CAM came from persons in the intimate circle of the patient and not from direct advertising. Remarkably, the age of our patients (50 years on average) might not be the prime target of e-commerce strategies, and we cannot rule out the influence of recommendations of younger family and friends as a result of digital media advertising. Indeed, we found that patients with relatives with RD use CAM more frequently. In line with this, YouTube and various social networks, including Facebook, have been shown to have an increasing number of videos containing medical information, or even posts geared toward the specific use of CAM, which can spread inaccuracies due to lack of scientific rigor [48,49].

Regarding rheumatologists' opinion about CAM, a systematic review showed a trend to favor some forms of CAM in other specialties [50]. Rheumatologists report personal use of CAM in 34% of patients, with psychosocial support and exercise being the most favored. In our study, the rheumatologist's attitude toward the use of CAM did not differ between disagreement, agree, or indifferent. This range of reactions from the rheumatologist can be partially explained by the great variety of CAM types. The use of CAM and the preconceptions around it differ from those of physicians in different countries and specialties. General physician show a more favorable attitude toward CAM than more specialized physician [51–53]. A more positive perception of the physician is also influenced by being female or younger.

Although CAM is a defined and uniform concept [2]; in the real world, CAM is composed of a heterogeneous group of treatments. Likely, the CAM concept is an oversimplification, which includes an undefinable moiety of traditional and novel allegedly beneficial treatments. In Mexico, the regulation of CAM is deficient, and several treatments, either herbal or chemically produced, have no formal and legally unattainable distribution routes and sites; indeed, their real content is unknown, and their manufacturing processes are far from accountable [54,55].

It is worth reconsidering the WHO position inclusiveness to CAM and its potential influence in its use. By endorsing the use of a great variety of therapeutic modalities, the WHO creates a double standard that equates unproven therapies at the same level of standard scientific treatments given their wide availability to purchase legally. If we accept this, we deny (at least partially) the role of scientific work and the need for scientific evidence in establishing therapeutics. In the era of the coronavirus disease pandemic, we have witnessed a widespread resistance to both scientific knowledge and lack of respect and recognition of the indications of formal authorities [56–60], including the WHO. In particular, with the widespread use of social media, the voices of unprepared, unaccountable influencers successfully challenge the official statements from highly trained health professionals [58–60]. We are likely to reconsider openness and return to the rigor of evidence as a supreme choice determinant in many aspects of public life.

Our study has several limitations. Although our sample size is comparable to that of many previous studies, it did not compare different RDs because the number of patients varied significantly. Moreover, due the study design, the responses of the patients partially depended on their memory. Therefore, the use of CAM in proportion and its variety may be underestimated. In addition, despite our best efforts, the CAM visual catalog might have been non-exhaustive. Additionally, despite some attempts, we were unable to prove the presence of cannabinoids in the ointments that alleged to contain them.

In conclusion, CAM use is frequent in our population of patients with RD them reporting a limited improvement in their well-being. In some cases, CAM was used due to the patient's worsening condition, either as a result of insufficient benefit from the formal treatment or limited access to an effective therapy; in this scenario, the patients search in their surroundings for alternatives that could alleviate their complaints or improve their quality of life. The increased availability of CAM and a distortedly advertised benefit/risk ratio repeatedly attract patients to try new modalities despite little or very limited benefit.

## Supporting information

**S1 File.**
(SAV)

**S2 File.**
(DOCX)

## Author Contributions

**Conceptualization:** César Pacheco-Tena.

**Data curation:** Susana Aideé González-Chávez.

**Formal analysis:** Susana Aideé González-Chávez, César Pacheco-Tena.

**Funding acquisition:** César Pacheco-Tena.

**Investigation:** Claudia Isabel Caballero-Hernández, César Pacheco-Tena.

**Methodology:** Ingris Peláez-Ballestas, Everardo Álvarez-Hernández, César Pacheco-Tena.

**Project administration:** César Pacheco-Tena.

**Resources:** Adelfia Urenda-Quezada, Greta Cristina Reyes-Cordero, César Pacheco-Tena.

**Supervision:** Susana Aideé González-Chávez, César Pacheco-Tena.

**Validation:** Adelfia Urenda-Quezada, Greta Cristina Reyes-Cordero, Ingris Peláez-Ballestas, Everardo Álvarez-Hernández.

**Visualization:** Claudia Isabel Caballero-Hernández, Susana Aideé González-Chávez, César Pacheco-Tena.

**Writing – original draft:** Claudia Isabel Caballero-Hernández, Susana Aideé González-Chávez, César Pacheco-Tena.

**Writing – review & editing:** Adelfia Urenda-Quezada, Greta Cristina Reyes-Cordero, Ingris Peláez-Ballestas, Everardo Álvarez-Hernández.

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
