## [Decision Letter · Decision Letter 0]

4 May 2021

PONE-D-21-07509

High prevalence of Complementary and Alternative Medicine and limited perceived efficacy in patients with Rheumatic Diseases in Mexico: Cross-sectional Study.

PLOS ONE

Dear Dr. Pacheco-Tena,

Thank you for submitting your manuscript to PLOS ONE. After careful consideration, we feel that it has merit but does not fully meet PLOS ONE’s publication criteria as it currently stands. Therefore, we invite you to submit a revised version of the manuscript that addresses the points raised during the review process.

We look forward to receiving your revised manuscript.

Kind regards,

Jenny Wilkinson, PhD

Academic Editor

PLOS ONE

Journal Requirements:

2. Please include a separate caption for each figure in your manuscript.

Additional Editor Comments:

Thank you for your submission, reviewers have provided suggestions for areas to improve your manuscript particularly around the description of the methods you used. I encourage you to consider these comments and provide revisions to your work.

Reviewers' comments:

Reviewer's Responses to Questions

**Comments to the Author**

1. Is the manuscript technically sound, and do the data support the conclusions?

Reviewer #1: Yes

Reviewer #2: Partly

2. Has the statistical analysis been performed appropriately and rigorously? 

Reviewer #1: Yes

Reviewer #2: Yes

3. Have the authors made all data underlying the findings in their manuscript fully available?

Reviewer #1: No

Reviewer #2: Yes

4. Is the manuscript presented in an intelligible fashion and written in standard English?

Reviewer #1: Yes

Reviewer #2: No

5. Review Comments to the Author

Reviewer #1: Review

Dear authors

Thank you so much for a very interesting research entitles (High prevalence of Complementary and Alternative Medicine and limited perceived efficacy in patients with Rheumatic Diseases in Mexico: Cross-sectional Study) I enjoyed reading the paper

Kindly address the following

Title: is guiding to the results with the word high, suggest remove

Introduction

• In the end of introduction, you stated that CAM use in RA was reported by ref 24, suggest adding more information or at least the reported prevalence

Methods

• Was STOBE guidelines used for methodology?

• In what language was the survey

• Were all the tools validated in target language

• Was the final survey at least subject to face validity before use?

• The statement is misleading suggest to rephrase ‘The data was coded by three researchers (ICH, SGC, CPT) defining dichotomy to ease the logistic regression based on the average’

• Is a standard well-known definition available for this classification ‘high mechanical demand (blue-collar workers) or low mechanical demand (white-collar workers)’ if yes please provide reference

• In regression you used ‘univariate’ but actually it was binary logistic regression

• From what I understand we use cut off of <0.05 as significant ‘Significant variables (p-value of < 0.2) in the univariate analysis were included in the multivariable logistic regression analyses.

• I didn’t see the definition of CAM

Results

The diagnosis of RD was it mentioned in patient file or reported by patient

Time frame of recruitment wasn’t mentioned

Was the survey RAPID approved for assessing disease activity in all RD and not to RA

I think the pilot part should be moved to methods as part of tools and measurements

Even the establishment of visual catalogue should be moved to methods

Discussion

Too long suggest reduce and keep it focused on results that were found

Figures

Not clear

Reviewer #2: This manuscript contains interesting results which I believe should be published. The quality of the paper, however, is lacking. After reading the abstract, I was looking forward to reading the entire paper.

The introduction section needs to be re-written. I believe the manuscript would benefit from the introduction of standard definitions of CAM upfront (i.e., paragraph #1). Use standard definitions of complementary medicine and alternative medicine to provide a better framework for this paper. For example, you might consider utilizing WHO or NCCIH definitions and then please cite.

In the last sentence, of the second paragraph, of the introduction section you make a statement about CAM’s lack of efficacy. I believe this statement to be inaccurate as the majority of CAM modalities currently lack evidence of efficacy. Either way such a statement needs to be cited. The rationale for the study is difficult to follow. Overall, the introduction section needs to provide a more coherent justification for the study. Please watch run-on sentences and typographical errors throughout the introduction section.

The methods section contains the majority of the necessary information. It is easy for the reader to get lost in this section of the paper unfortunately. Perhaps, better organization including sub-headings would help here, (ex. survey instrument, pilot study, sample size, data analysis, etc.). Again, watch run-on sentences and typographical errors.

In the results section you refer to this as an “interview”. Was this a “survey” or “interview”? I am assuming you are referring to survey methodology, please state in the methods section how the survey/questionnaire was distributed (pen and paper, web-based, etc.). This needs to be clarified-as is, the study could not be replicated. Paragraph #6 of the results section has a “discussion section” feel to it-minor word changes would remedy. The cannabis related results are interesting. I think readers would find these results important science, especially with the current interest in cannabis research in the scientific community. Again, watch run-on sentences and typographical errors throughout. The second paragraph, of the results, needs to be written more clearly.

The discussion section does not support the authors’ conclusions in the final paragraph of the manuscript. In my opinion, these conclusions are too broad. I do believe this paper is the culmination of a study with interesting results of a smaller scale. It is true the results point to some public health issues regarding CAM, but the quality of the paper is disappointing and the text very difficult for the reader to follow. Based on the fact that PLOS ONE does not copyedit manuscripts this paper must be rejected. I do hope the authors consider re-writing because it does appear this study contains interesting science and important results. Again, watch run-on sentences and typographical errors throughout. Also, when discussing the limitations of the study don’t refer to the study as a retrospective study-it is cross-sectional, as you stated in the methods. The “retrospective” part you are referring to is recall bias.

6. PLOS authors have the option to publish the peer review history of their article (what does this mean?). If published, this will include your full peer review and any attached files.

Reviewer #1: No

Reviewer #2: No

---

## [Author Response · Author response to Decision Letter 0]

28 Jun 2021

From: César Pacheco-Tena, M.D., Ph.D.

PABIOM Laboratory

Facultad de Medicina y Ciencias Biomédicas, Universidad Autónoma de Chihuahua.

Circuito Universitario, Campus II, Chihuahua, Chih., México. C.P. 31125.

Tel: (52) 614-2386030 ext. 3586

e-mail: dr.cesarpacheco@gmail.com

June 16, 2021

To: Editor and Reviewers

PLOS ONE

Re: Response to Reviewers

Dear Editor

I am pleased to resubmit for publication the revised version of the manuscript “Prevalence of Complementary and Alternative Medicine despite limited perceived efficacy in patients with Rheumatic Diseases in Mexico: Cross-sectional Study”. We appreciate your constructive criticism and that of the reviewers. We reviewed our manuscript in order to respond to the questions raised by the reviewers and hope that now it can be judged as acceptable for publication in your prestigious journal. Please find below a response “point by point” of the questions and criticisms of the referees.

Reviewer#1: Review

Dear authors

Thank you so much for a very interesting research entitles (High prevalence of Complementary and Alternative Medicine and limited perceived efficacy in patients with Rheumatic Diseases in Mexico: Cross-sectional Study) I enjoyed reading the paper

Kindly address the following

1. Title: is guiding to the results with the word high, suggest remove

The word "high" was removed from the title. 

2. Introduction: In the end of introduction, you stated that CAM use in RA was reported by ref 24, suggest adding more information or at least the reported prevalence.

More information was described from the study that evaluated the association of significant risk perception with the use of CAM in Hispanic patients with RA. The prevalence of use, the prevalence of significant risk perceived, and the factors associated with the use of CAM were included.

The prevalence of CAM use in Mexican patients with RD was referenced in the numbers 22 to 26 as mentioned in the beginning of the paragraph . 

3. Methods: Was STROBE guidelines used for methodology?

Yes, the STROBE-cross-sectional study was applied. 

4. In what language was the survey

The Spanish language was used for the interviews, also the questionnaires were in Spanish. This information has already been added in the methods section. The patients did not answer questionnaires, everything was asked in the interview directly by the interviewer.

5. Were all the tools validated in target language

Yes, we have added the references to the text.

6. Was the final survey at least subject to face validity before use?

Yes, the patients in the construction stage were interviewed in regard to the images, the patients only looked at the images to recall the use of products. This information was now included in the methods sections ("instruments" and "pilot test").

7. The statement is misleading suggest to rephrase ‘The data was coded by three researchers (ICH, SGC, CPT) defining dichotomy to ease the logistic regression based on the average’

The paragraph was rephrased.

8. Is a standard well-known definition available for this classification ‘high mechanical demand (blue-collar workers) or low mechanical demand (white-collar workers)’ if yes please provide reference

Yes, there is a definition. The Cambridge Dictionary (https://dictionary.cambridge.org/) defines:

Blue-collar workers do work needing strength or physical skill rather than office work.

White-collar: relating to people who work in offices, doing work that needs mental rather than physical effort.

On the other hand, more than 500 scientific articles from PubMed database have been published using these terms to differentiate workers depending on the workload they perform (PubMed search: white-collar blue-collar worker).

9. In regression you used ‘univariate’ but actually it was binary logistic regression

The term "univariate analysis" was changed to "binary logistic regression" throughout the text

10. From what I understand we use a cut off of <0.05 as significant ‘Significant variables (p-value of < 0.2) in the univariate analysis were included in the multivariable logistic regression analyses.

Variables resulting from the binary analysis with p <0.2 were used for the multivariate logistic regression. These variables were not statistically significant according to our criteria (p <0.05), so the word "significant" was eliminated from the sentence for clarity.

11. I didn’t see the definition of CAM

The definition of CAM by the WHO and the NCCAM was added in the first paragraph of the introduction.

12. Results: The diagnosis of RD was it mentioned in patient file or reported by patient

All the patients were previously known and diagnosed by the referring physicians (CPT, GRC, AUQ) and were under treatment and follow-up, there were no external referrals. This information was included in the methods section ("participants").

13. Time frame of recruitment wasn’t mentioned

The time frame of the study was now added in the methods section

14. Was the survey RAPID approved for assessing disease activity in all RD and not to RA

Yes, the RAPID questionnaire was used in patients with RD other than RA, although it was originally designed for RA. RAPID-3 has proved to be reliable and applicable to patients with RD other than RA, including SLE and Ax-SpA. These three diseases (RA, SLE and A-SpA) conform over 90% of our sample, and the use of a single functional measure allows us to include functional status as in the overall statistical analysis. The references of its validation in SLE and SpA were added in the methods section. 

15. I think the pilot part should be moved to methods as part of tools and measurements. Even the establishment of a visual catalogue should be moved to methods.

In the methods section, the construction of the visual CAM catalog and the pilot have been included and detailed.

16. Discussion: Too long suggest reduce and keep it focused on results that were found

The discussion has been reduced and we try to focus more on our results. Thanks for your valuable suggestion.

17. Figures: Not clear

We have now included the figure legend as we omitted it in the initial submission. We hope it is clearer now.

Reviewer #2: This manuscript contains interesting results which I believe should be published. The quality of the paper, however, is lacking. After reading the abstract, I was looking forward to reading the entire paper.

1. The introduction section needs to be re-written. I believe the manuscript would benefit from the introduction of standard definitions of CAM upfront (i.e., paragraph #1). Use standard definitions of complementary medicine and alternative medicine to provide a better framework for this paper. For example, you might consider utilizing WHO or NCCIH definitions and then please cite. 

The definition of CAM by the WHO and the NCCAM was added in the first paragraph of the introduction.

2. In the last sentence, of the second paragraph, of the introduction section you make a statement about CAM’s lack of efficacy. I believe this statement to be inaccurate as the majority of CAM modalities currently lack evidence of efficacy. Either way such a statement needs to be cited. 

The sentence was rephrased and more supporting references were added.

3. The rationale for the study is difficult to follow. Overall, the introduction section needs to provide a more coherent justification for the study.

Thanks for your comment, we modified the introduction and we hope the purpose and justification is more clear for the reader.

4. Please watch run-on sentences and typographical errors throughout the introduction section. 

The text was revised and edited by an English language editing service.

5. The methods section contains the majority of the necessary information. It is easy for the reader to get lost in this section of the paper unfortunately. Perhaps, better organization including subheadings would help here, (ex. survey instrument, pilot study, sample size, data analysis, etc.). 

The methods section was now organized into sections to make it easier to read: 

➔ Study design

➔ Participants

➔ Variables and Interview instruments

➔ Pilot study

➔ Sample size

➔ Statistical analysis 

➔ Ethical approval.

The sections were also rewritten with greater precision to increase clarity.

6. Again, watch run-on sentences and typographical errors. 

The text was revised and edited by an English language editing service.

7. In the results section you refer to this as an “interview”. Was this a “survey” or “interview”? I am assuming you are referring to survey methodology, please state in the methods section how the survey/questionnaire was distributed (pen and paper, web-based, etc.). This needs to be clarified-as is, the study could not be replicated. 

Thank you for your comment, indeed our text contained heterogeneity between the terms interview and survey. The data collection strategy was through interviews, the patients did not fill out any questionnaire themselves (neither by hand nor electronically). The interviewer asked the questions and the patient answered them. The interviewer also filled out the questionnaires with the patient's responses. Now the terms are homogeneous in the text specifying the use of interviews.

8. Paragraph #6 of the results section has a “discussion section” feel to it-minor word changes would remedy. 

The paragraph was revised and wording modifications were made to state only the findings.

9. The cannabis related results are interesting. I think readers would find these results important science, especially with the current interest in cannabis research in the scientific community.

We have included comments both in the results and the discussion, the commonest use of cannabis is in an ointment, no significant use of oral or smoked cannabis products is referred, no synthetic cannabinoids appeared as a trend, therefore is an interesting finding but mostly explained by its wide availability. 

10. Again, watch run-on sentences and typographical errors throughout. 

The text was revised and edited by an English language editing service.

11. The second paragraph, of the results, needs to be written more clearly.

The paragraph was rewritten for more clarity.

12. The discussion section does not support the authors’ conclusions in the final paragraph of the manuscript. In my opinion, these conclusions are too broad.

We thank you for our comment, the conclusion has been limited to what can be actually concluded given our results. We hope you agree

13. I do believe this paper is the culmination of a study with interesting results of a smaller scale. It is true the results point to some public health issues regarding CAM, but the quality of the paper is disappointing and the text very difficult for the reader to follow. Based on the fact that PLOS ONE does not copy edit manuscripts this paper must be rejected. I do hope the authors consider re-writing because it does appear this study contains interesting science and important results.

We hope our re-writing and editing process had improved the content and make it easier to be followed and understood. 

14. Again, watch run-on sentences and typographical errors throughout.

The text was revised and edited by an English language editing service.

15. Also, when discussing the limitations of the study don’t refer to the study as a retrospective study-it is cross-sectional, as you stated in the methods. The “retrospective” part you are referring to is recall bias.

The word “retrospective” was removed

---

## [Decision Letter · Decision Letter 1]

2 Aug 2021

PONE-D-21-07509R1

Prevalence of Complementary and Alternative Medicine despite limited perceived efficacy in patients with Rheumatic Diseases in Mexico: Cross-sectional Study

PLOS ONE

Dear Dr. Pacheco-Tena,

Thank you for submitting your manuscript to PLOS ONE. After careful consideration, we feel that it has merit but does not fully meet PLOS ONE’s publication criteria as it currently stands. Therefore, we invite you to submit a revised version of the manuscript that addresses the points raised during the review process.

We look forward to receiving your revised manuscript.

Kind regards,

Jenny Wilkinson, PhD

Academic Editor

PLOS ONE

Journal Requirements:

Additional Editor Comments (if provided):

Thank you for your submission, the revisions have significantly improved the work. Reviewer comments on your revisions are provided and highlight that the Discussion would benefit from some further revision.

Reviewers' comments:

Reviewer's Responses to Questions

**Comments to the Author**

1. If the authors have adequately addressed your comments raised in a previous round of review and you feel that this manuscript is now acceptable for publication, you may indicate that here to bypass the “Comments to the Author” section, enter your conflict of interest statement in the “Confidential to Editor” section, and submit your "Accept" recommendation.

Reviewer #2: (No Response)

2. Is the manuscript technically sound, and do the data support the conclusions?

Reviewer #2: Partly

3. Has the statistical analysis been performed appropriately and rigorously? 

Reviewer #2: Yes

4. Have the authors made all data underlying the findings in their manuscript fully available?

Reviewer #2: Yes

5. Is the manuscript presented in an intelligible fashion and written in standard English?

Reviewer #2: No

6. Review Comments to the Author

Reviewer #2: The following constitutes a review of the revised manuscript: Prevalence of complementary and alternative medicine despite limited perceived efficacy in patients with rheumatic diseases in Mexico: Cross-sectional study. The authors have obviously put effort into revising this paper-Thank you!

ABSTRACT:

*As it reads "The prevalence of CAM use was reported by 59.2% of patients who informed a total of 155 different therapies." is confusing. I suggest re-writing to the following: The prevalence of CAM use was reported by 59.2% of patients, which informed a total of 155 different therapies.

*Watch for Typos

INTRODUCTION:

*2nd Paragraph, last sentence - The way it reads is confusing. I think you are trying to say the following: ...of those CAM therapies that have been tested many lack comprehensive testing and few have failed to prove efficacy. In my onion, this should be re-written for clarity.

*Overall, the introduction section is much better. The majority of the information is contained in this section, but it still needs to flow better to support the stated objective: "Describe the prevalence, diversity, and factors related to CAM use in RD patients.

*Watch for Typos

METHODS:

*The reorganization of this section makes it much easier to follow - Nice Job!

*A few Typos

RESULTS:

* "very variable" (2nd to last paragraph)-redundant.

DISCUSSION:

*At the end of paragraph three and into paragraph four the reader is likely to get lost.

*Paragraphs five and six: Again, this is difficult to follow and there are no citations to back the statements.

*Limitations section: Recall bias may over- or under-estimate some value. The idea is, what the respondent tells the interviewer isn't the "truth".

*This is a descriptive study of the prevalence, diversity, patient perceptions, and factors related to CAM use in a specific population of RD patients. Don't make conclusions and take the discussion beyond what your study supports. Your objectives are good and are supported by data and results. This has a place in the scientific literature - in my opinion. Please stick to your findings in the discussion section to strengthen this manuscript.

7. PLOS authors have the option to publish the peer review history of their article (what does this mean?). If published, this will include your full peer review and any attached files.

Reviewer #2: No

---

## [Author Response · Author response to Decision Letter 1]

18 Aug 2021

From: César Pacheco-Tena, M.D., Ph.D.

PABIOM Laboratory

Facultad de Medicina y Ciencias Biomédicas, Universidad Autónoma de Chihuahua.

Circuito Universitario, Campus II, Chihuahua, Chih., México. C.P. 31125.

Tel: (52) 614-2386030 ext. 3586

e-mail: dr.cesarpacheco@gmail.com

August 18, 2021

To: Editor and Reviewers

PLOS ONE

Re: Response to Reviewers

Dear Editor

I am pleased to resubmit for publication the revised version 2 of the manuscript “Prevalence of Complementary and Alternative Medicine despite limited perceived efficacy in patients with Rheumatic Diseases in Mexico: Cross-sectional Study”. We appreciate your constructive criticism and that of the reviewers. We reviewed our manuscript in order to respond to the questions raised by the reviewers and hope that now it can be judged as acceptable for publication in your prestigious journal. Please find below a response “point by point” of the questions and criticisms of the referees.

Reviewer #2: The following constitutes a review of the revised manuscript: Prevalence of complementary and alternative medicine despite limited perceived efficacy in patients with rheumatic diseases in Mexico: Cross-sectional study. The authors have obviously put effort into revising this paper-Thank you!

1. ABSTRACT:

*As it reads "The prevalence of CAM use was reported by 59.2% of patients who informed a total of 155 different therapies." is confusing. I suggest re-writing to the following: The prevalence of CAM use was reported by 59.2% of patients, which informed a total of 155 different therapies.

* Watch for Typos

Thanks for your suggestion, we have made the change.

* Watch for Typos

The text was previously reviewed and edited by a service specialized in editing the English language. Additionally, we have made an intentional search for typos by ourselves and with the use of software.

2. INTRODUCTION: 

*2nd Paragraph, last sentence - The way it reads is confusing. I think you are trying to say the following: ...of those CAM therapies that have been tested many lack comprehensive testing and few have failed to prove efficacy. In my onion, this should be re-written for clarity

Thanks for your suggestion, we have made the change.

*Overall, the introduction section is much better. The majority of the information is contained in this section, but it still needs to flow better to support the stated objective: "Describe the prevalence, diversity, and factors related to CAM use in RD patients.

Thank you for your encouraging comment, we have now added some perspective in regard to the need to explore local scenarios in regard of CAM use to better control the use of risky alternatives and also to drive the discussion with the patients regarding compliance to standards of care; we needed to explore the local dynamics of CAM use to update the Mexican perspective and to define our situation compared to other populations. We hope this makes a clearer justification for the objective. 

* Watch for Typos

The text was previously reviewed and edited by a service specialized in editing the English language. Additionally, we have made an intentional search for typos by ourselves and with the use of software.

3. METHODS:

*The reorganization of this section makes it much easier to follow - Nice Job!

Thank you

*A few Typos

The text was previously reviewed and edited by a service specialized in editing the English language. Additionally, we have made an intentional search for typos by ourselves and with the use of software.

4. RESULTS:

* "very variable" (2nd to last paragraph)-redundant.

Thank you, we have corrected this sentence

5. DISCUSSION:

*At the end of paragraph three and into paragraph four the reader is likely to get lost.

We have changed the wording of these paragraphs so that they have a better understanding of our ideas.

*Paragraphs five and six: Again, this is difficult to follow and there are no citations to back the statements.

We have worked on improving the wording of the discussion to make it easier to follow, we have also added quotes that support our ideas.

*Limitations section: Recall bias may over- or under-estimate some value. The idea is, what the respondent tells the interviewer isn't the "truth".

We have erased the term and pointed that the limitation is the retrospective nature of the information recollection

*This is a descriptive study of the prevalence, diversity, patient perceptions, and factors related to CAM use in a specific population of RD patients. Don't make conclusions and take the discussion beyond what your study supports. Your objectives are good and are supported by data and results. This has a place in the scientific literature - in my opinion. Please stick to your findings in the discussion section to strengthen this manuscript.

We have made an effort to limit our discussion and conclusions to those supported by our data; We now believe the discussion is more concrete and clearer, we hope you like it.

---

## [Editor Report · Decision Letter 2]

31 Aug 2021

Prevalence of Complementary and Alternative Medicine despite limited perceived efficacy in patients with Rheumatic Diseases in Mexico: Cross-sectional Study

PONE-D-21-07509R2

Dear Dr. Pacheco-Tena,

We’re pleased to inform you that your manuscript has been judged scientifically suitable for publication and will be formally accepted for publication once it meets all outstanding technical requirements.

Kind regards,

Jenny Wilkinson, PhD

Academic Editor

PLOS ONE
---

## [Editor Report · Acceptance letter]

17 Sep 2021

PONE-D-21-07509R2 

Prevalence of Complementary and Alternative Medicine despite limited perceived efficacy in patients with Rheumatic Diseases in Mexico: Cross-sectional Study 

Dear Dr. Pacheco-Tena:

I'm pleased to inform you that your manuscript has been deemed suitable for publication in PLOS ONE. Congratulations! Your manuscript is now with our production department. 

Kind regards, 

on behalf of

Dr Jenny Wilkinson 

Academic Editor

PLOS ONE